# The Number of Topics Optimization: Clustering Approach

**Fedor Krasnov [1],\* and Anastasiia Sen [2]**

[1]    Gazpromneft STC, 75-79 Moika River Emb., 190000 Saint Petersburg, Russia
[2]    Faculty of Applied Mathematics and Control Processes, Saint Petersburg State University,
      7-9 Universitetskaya Emb., 199034 Saint Petersburg, Russia; anastasiia.sen@gmail.com
\*    Correspondence: krasnov.fv@gazprom-neft.ru

**Abstract:**   Although topic models have been used to build clusters of documents for more than ten years, there is still a problem of choosing the optimal number of topics. The authors analyzed many fundamental studies undertaken on the subject in recent years. The main problem is the lack of a stable metric of the quality of topics obtained during the construction of the topic model. The authors analyzed the internal metrics of the topic model: coherence, contrast, and purity to determine the optimal number of topics and concluded that they are not applicable to solve this problem. The authors analyzed the approach to choosing the optimal number of topics based on the quality of the clusters. For this purpose, the authors considered the behavior of the cluster validation metrics: the Davies Bouldin index, the silhouette coefficient, and the Calinski-Harabaz index. A new method for determining the optimal number of topics proposed in this paper is based on the following principles: (1) Setting up a topic model with additive regularization (ARTM) to separate noise topics; (2) Using dense vector representation (GloVe, FastText, Word2Vec); (3) Using a cosine measure for the distance in cluster metric that works better than Euclidean distance on vectors with large dimensions. The methodology developed by the authors for obtaining the optimal number of topics was tested on the collection of scientific articles from the OnePetro library, selected by specific themes. The experiment showed that the method proposed by the authors allows assessing the optimal number of topics for the topic model built on a small collection of English documents.

**Keywords:** clustering; additive regularization topic model; validation metrics; Davies Bouldin Index; ARTM

## 1. Introduction

Topic models have been used successfully for clustering texts for many years. One of the most common approaches to topic modeling is the Latent Dirichlet Allocation (LDA) [1]. It models a fixed number of topics which selected as a parameter based on the Dirichlet distribution for words and documents. The result is a flat, soft probabilistic clustering of terms by topics and documents by topics. All the topics received are equal, they do not create any characteristic signs that could help the researcher to identify the most useful topics, that is, to choose a subset of topics that are best suited for human interpretation. The problem of finding the metric characterizing such interpretability is a subject of study by many researchers [2–5].

The topic model is not able to read the insights of the researcher and therefore must have the settings for the task that the researcher is going to solve. According to studies [6,7] topic models based on the LDA have the following parameters:

- $\alpha$: The parameter of the prior Dirichlet distribution for "documents-topics";
- $\beta$: Parameter of the prior Dirichlet distribution for "topics-words";
- $tn$: The number of topics;
- $b$: The number of discarded initial iterations according to Gibbs sampling;
- $n$: The number of samples;
- $si$: Sampling interval.

In the recent study [7], published in 2018, an attempt was made to find the optimal values of the above parameters using the algorithm of differential evolution [8]. The authors chose a modified Jaccard similarity metric as the cost-function. As a result, a new Latent Dirichlet Allocation Differential Evolution (LDADE) algorithm was created, in which free parameters from the differential evolution algorithm appeared and they also need to be optimized.

There is a difference between evaluating of a complete set of topics and evaluating individual topics to filter out unwanted information (noise). To evaluate a complete set of topics, researchers usually look at the perplexity [9] for the corpus of documents.

This approach does not work very well according to the results of studies [10,11] because the perplexity does not have an absolute minimum, and with increasing iterations, it becomes asymptotic [12].

The most common use of perplexity is to detect the "elbow effect", that is, when the pattern of growth in the orderliness of the model changes drastically. Perplexity depends on the power of the dictionary and the frequency distribution of words in the collection, hence we get its drawbacks:

- It cannot evaluate the quality of deletion of stop words and non-topic words;
- It cannot compare rarefying methods for dictionary;
- It cannot compare uni-gram and n-gram models.

The authors of the LDA made a study of the quality of topics using the Bayesian approach in [13]. It is important to note that the Hierarchical Dirichlet process (HDP) [14] solves the problem of the optimal number of topics for the whole collection, but not for a specific document.

Let us pay attention to the difference between the LDA, HDP, and hierarchical Latent Dirichlet Allocation (hLDA) [15,16], since these are different topic models. LDA creates a flat, soft probabilistic clustering of terms by topic and documents by topic. In the HDP model, instead of a fixed number of topics for a document, the Dirichlet process generates the number of topics, which leads to the fact that the number of topics is also a random variable. The "hierarchical" part of the name belongs to another level added by the Dirichlet process, which creates several topics, and the topics themselves are still flat clusters. The hLDA model is an adaptation of the LDA, which models the topics as the distribution of a new, predetermined number of topics taken from the Dirichlet distribution. The hLDA model still considers the number of topics as a hyper parameter, that is, regardless of the data. The difference is that clustering is now hierarchical: The hLDA model studies the clustering of the first set of topics, providing more general abstract relationships between topics (and, therefore, words and documents). Note that all three models described (LDA, HDP, hLDA) add a new set of parameters that require optimization, as is noted in the study [17].

One of the main requirements for topic models is human interpretability [18]. In other words, whether the topics contain words that, according to a person's subjective judgments, are representative of a single

coherent concept. In [19], Newman showed that the human assessment of interpretability well correlates with an automated quality measure called coherence.

Recent research [20] proposed to minimize the Rényi and Tsallis entropies to find the optimal number of topics in the topic modeling. In this study, topic models derived from large collections of texts are considered as non-equilibrium complex systems, where the number of topics is considered as the equivalent of temperature. This allows us to calculate the free energy of such systems—the value through which the Renyi and Tsallis entropies are easily expressed. The metrics obtained based on entropy make it possible to find a minimum depending on the number of topics for large collections, but in practice we rarely find small collections of documents.

A study [21] proposed a matrix approach to improving the accuracy of determining topics without using optimization. On the other hand, the study [22] noted that increasing the accuracy of the model is contrary to human interpretability. In particular, a more recent study [23] created the VisArgue framework designed to visualize the model's learning process to determine the most explainable topics.

The use of the statistical measure of term frequency (TF) divided by inverse document frequency (IDF) as a metric for quantifying the quality of topics was studied in [24]. There is also a series of studies combining the advantages of topic models and dense representations of word-vectors [25–28].

The motivation of the research conducted by the authors of this paper was the fact that the study of a stable metric for the quality of topics continues. Moreover, the use of cluster analysis is one of the tools for analyzing the stability of topics [29] and the optimal number of topics [30], but it does not consider the benefits of the special training capabilities of the topic model with sequential regularization and dense representation of word-vectors.

To validate the quality of clusters, many metrics have been developed. Among them, there are the partition coefficient [31], Dunn index [32], Davies Bouldin Index (DBI) [33] and its modifications [34,35], and silhouette coefficient [36]. Nevertheless, in the case of a topic model, we already have clusters of topics and do not need a clustering algorithm; we need only to evaluate the clusters obtained. For validation of clusters it is necessary to consider them in space with concepts of proximity and distance. For words, such a space is a vector representation of words. Significant results in this direction were obtained in previous research [37–39]. Words presented in the form of dense vectors, reflect the semantic representation and have the properties of proximity and distance. Therefore, presenting the topics in the form of dense vectors, the authors created a new variation of the Davies Bouldin Index metric for the topics, which the authors called *Cosine Davies Bouldin Index (cDBI)*.

The remainder of the paper is described as follows: The proposed methodology and research hypothesis are presented in Section 2; the results of testing a new quality metric are explained in Section 3. We conclude our paper in Section 4.

## 2. Research Methodology

Consider ways to build a topic model for a specific collection of documents. Collection is homogeneous if it contains documents of the same type. For example, a collection of scientific articles from one conference, created on a single template, is homogeneous. In the case of a homogeneous collection of scientific articles, each document has a similar structure, postulated by a conference template. All scientific articles consist of introduction, presentation of research results and conclusion. Thus, it is possible to present a document in the form of a distribution of the main topic and auxiliary topics: introduction and conclusion.

Of course, the main topics in different documents may be different. However, the collection of scientific articles may be limited to the choice of certain headings from the thematic rubrics of the

conference. Then the number of topics will be known. Figure 1 shows the matrix distribution of topics on the documents.

As we see on the left side of Figure 1, topic model leads to the emphasis of topics and their distribution homogeneously over the documents. Such a picture of the probabilities of the "topics-documents" matrix can be obtained using, e.g., models based on the LDA algorithm [1]. In addition, the right side of Figure 1 shows the result of the model with a sequential Additive Regularization of Topic Models (ARTM) [2]. The main and auxiliary topics are highlighted through the management of the learning process of the model. The principle of classifying a topic as auxiliary may be formulated as the existence of such a topic in the overwhelming number of documents. That is, the probabilities of the auxiliary topics will be distributed uniformly and tightly across the documents. Furthermore, the main topic will be a sparse vector for each document, since each document is characterized by one main topic.

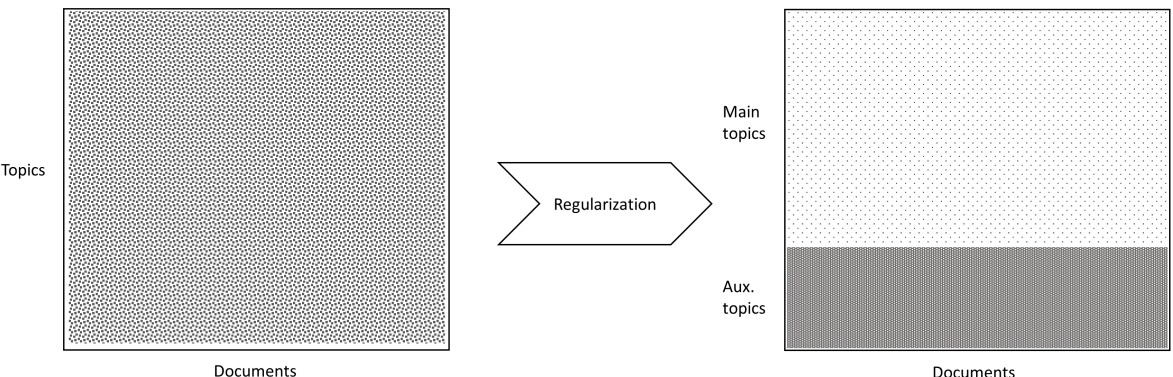

**Figure 1.** "Topics-documents" scheme.

We show that the existing internal metrics of the topic model are not suitable for determining the optimal number of topics. To do this, consider the internal automated metrics of the quality of topics. We introduce the concept of core topics:

$$W_t = \{w \in W \mid p\left(t|w\right) \geq threshold\} \, .$$

The following quality metrics of the topic model can be calculated based on the topics kernel:

○ Purity of the topics: $Purity = \sum_{w \in W_t} p(w|t)$
○ Size of the topic kernel: $|W_t|$
○ Contrast of the topics: $\frac{1}{|W_t|} \sum_{w \in W_t} p(t|w)$
○ Coherence of the topics: $Coh_t = \frac{2}{k(k-1)} \sum_{i=1}^{k-1} \sum_{j=1}^{k} PMI(w_i, w_j)$, where $k$ is the interval in which the combined use of words is calculated, point-wise mutual information $PMI(w_i, w_j) = \log \frac{N \cdot N_{w_i w_j}}{N_{w_i} \cdot N_{w_j}}$, $N_{w_i w_j}$—the number of documents in which words $w_i$ and $w_j$ appear in interval $k$ at least once. $N_{w_j}$—the number of documents in which the word $w_i$ appear at least once, and $N$ is the number of words in the dictionary.

As can be seen from the formulas for the internal metrics of the topic model, each of these metrics can be measured for a different number of topics ($tn$). Consider the behavior of the metric kernel size depending on the number of topics. With an increase in the number of topics, the core size will decrease, since the normalization conditions must be satisfied when constructing the matrices "topics-words" and

"documents-topics": The sum of the probabilities must be equal to one. For metrics, the purity of topics and the contrast of topics, the nature of the changes with an increase in the number of topics will also be monotonously decreasing, since the sum of the probabilities of the topics included in the core will decrease. On the other hand, for the metric, coherence to topics, behavior with an increase in the number of topics will be monotonously increasing, as the contribution from PMI will grow. The specific nature of the changes in the metrics examined may vary; therefore it is advisable to try to find the extreme point using numerical methods, if it is possible.

The quality of the topics of short messages from the point of view of clusters was reviewed in [40] using NMF (non-negative matrix factorization) and metrics reflecting the entropy of clusters. The matrix approach (Latent Semantic Indexing + Singular-Value Decomposition) to the selection of clusters of topics from the program code was investigated in [41] with a modified vector proximity metric. The research of the topic model's quality [30] use the metric of silhouette coefficient [36] with Euclidean distance for sparse subject vectors. Consequently, in these works, clusters in the space of dense vectors–words constituting topics and non-Euclidean distances in the metrics remain unexplored.

In [12,42,43], the instability of topics with respect to the order of processed documents was discovered and investigated. Therefore, to calculate the quality metrics of the topics, it is necessary to perform calculations for the corpus of documents with a random order to eliminate the dependence on the order of documents. The possibility of stabilizing the topic model with the help of regularization was shown in [44]. Based on the analysis, the authors formulated a methodological framework, depicted as a diagram in Figure 2.

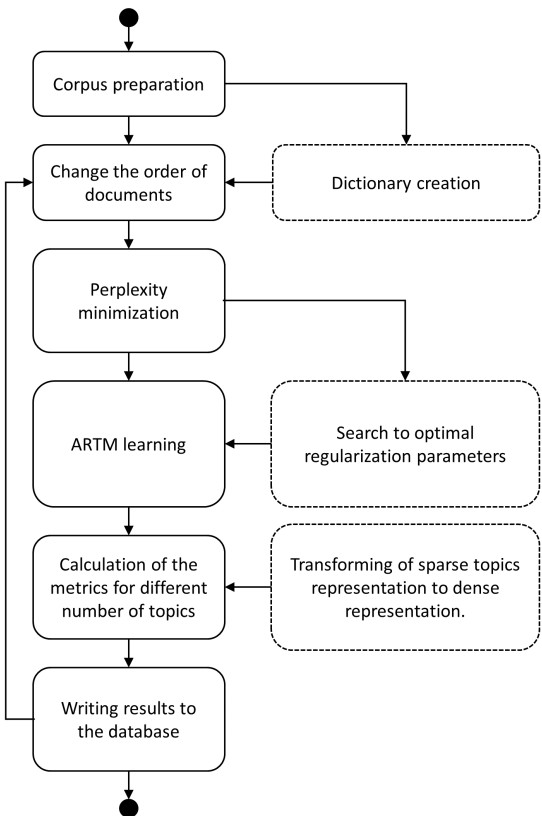

**Figure 2.** Research framework.

Figure 2 shows the sequence of actions repeated for one corpus of documents a significant number of times, in order comparable to the number of documents in the corpus. On the right, actions that are performed only once are displayed: The formation of a dictionary, the adjustment of the regularization parameters of the topic model, and the transformation of the sparse presentation space of topics into a dense representation. Based on this methodological framework, digital experiments were developed and carried out as described in the next section.

## 3. Experiment

For the experiment the corpus of scientific and technical articles on topics related to the development of oil and gas fields was chosen. In total, 1695 articles in English were selected in 10 areas of research according to the rubrics. The creation of a dictionary for the selected corpus is described in detail in the previous study by the authors [45]. To build a topic model, the BigARTM library was used, which allows for customization of the topic model by sequential regularization. The choice and adjustment of the regularization parameters of the topic model were made by the authors in a previous study [45]. To transform the sparse space of the vectors-words that make up the topics, the GloVe library was chosen [37]. To obtain a visual representation of the form of a dense representation of topics, a projection was made on a two-dimensional space with the distances preserved using the Multi-dimensional scaling (MDS) library [46]. Figure 3 presents the view of obtained clusters of topics.

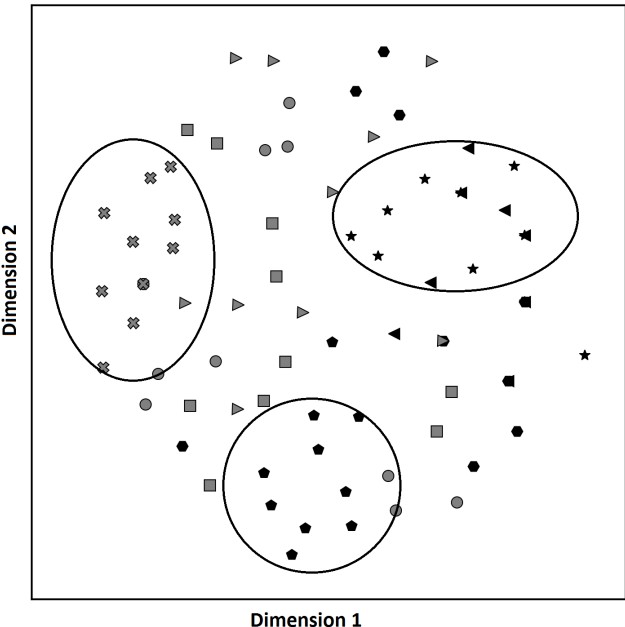

**Figure 3.** Projection of a dense presentation of topics with preservation of distances.

In Figure 3, two-dimensional projections of words from topics are highlighted with different markers. Ovals emphasize precise visual grouping of words in the topics.

Figure 4 presents the preliminary calculations of the main metrics behavior of the topic model, set up in accordance with the methodology proposed by the authors, depending on the number of topics.

As we can see from Figure 4, the nature of the dependencies is monotonous and does not allow determination of the optimal number of topics. Measurements of the main internal metrics are made for 1000 different random orders of documents. The y–axis represents the value of one standard deviation.

It is evident that for the metric the contrast of the core, the deviations are minimal. For the metrics, purity and coherence of the core, the greater values characterize the best quality of the topic model.

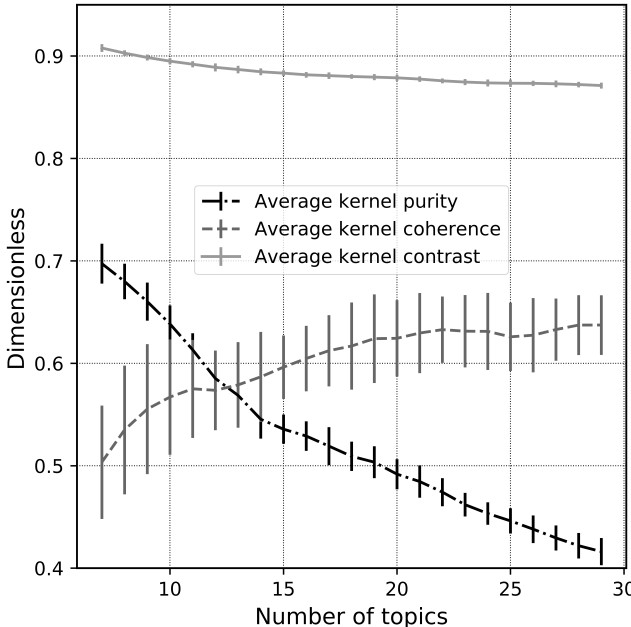

**Figure 4.** Dependencies of the main internal metrics of the quality of the topic model on the number of topics.

A characteristic point can be considered the number of topics equal to 12, when the curves of changes in the metric purity and coherence of topics intersect. Consider the dependencies of the following metrics: Calinski-Harabaz index [47], and silhouette coefficient [36], as used to validate the number of clusters.

According to Figure 5, the Calinski-Harabaz index and silhouette coefficient metrics do not make it possible to determine the optimal number of topics. As the number of topics increases, the values of these metrics decrease, which means that clusters become worse from the point of view of these metrics. The cDBI metric developed by the authors and shown in Figure 6 behaves differently depending on the number of topics.

In Figure 6 the maximum is clearly expressed when the number of topics is equal to 16. The algorithm for calculating the cDBI metric is based on the ideology of the Davies Bouldin index metric proposed in [33] and modified in [34,35].

---

**Algorithm 1:** Calculation of *Cosine Davies Bouldin Index (cDBI)* Metrics.

---

**Result:** $cDBI$

$V := GloVe(ARTM(tn, \mu, (corpus\ of\ texts)))$

**for** $t \in W :$ **do**

$\quad C_t := \sum_{i \in t} V_t^{(i)}$

$\quad D_t := \frac{1}{\dim t} \sum_{i \in t} \frac{C_t \cdot V_t^{(i)}}{|C_t| \cdot |V_t^{(i)}|}$

**end**

$cDBI := \frac{1}{\dim W} \sum_{t \in T} \frac{D_t}{C_t}$

---

In the above Algorithm 1, *T* denotes the number of selected, *μ*—this regularizing coefficients. Thus, using the cDBI metric, it is possible to find the optimal number of topics for a collection of documents.

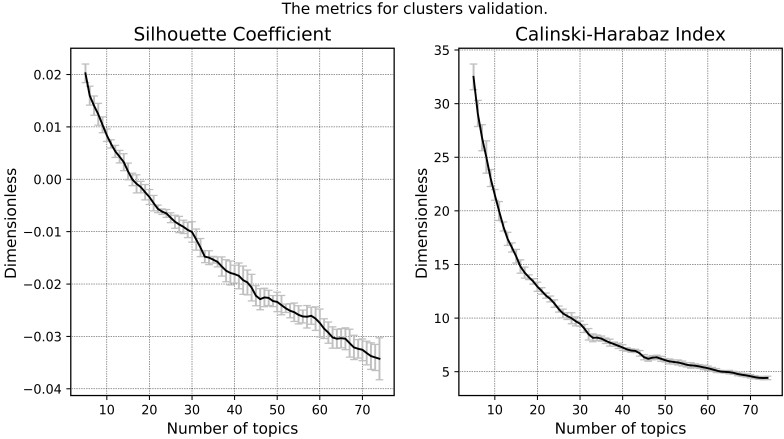

**Figure 5.** Cluster Validation Metrics.

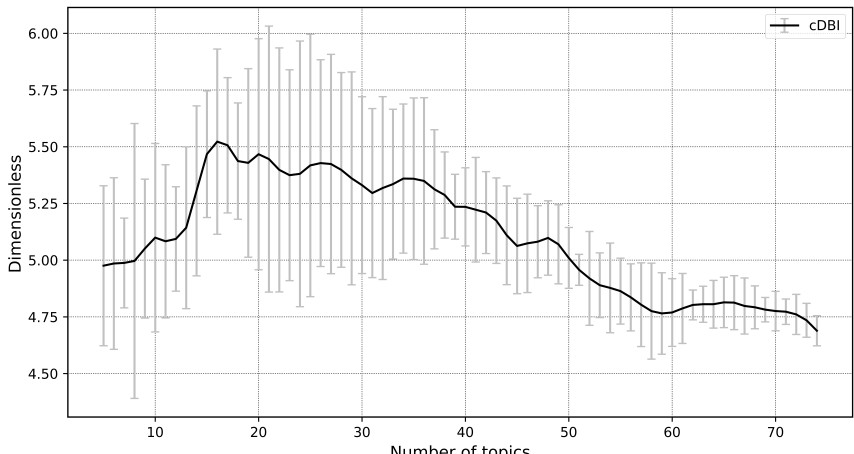

**Figure 6.** Cosine Davies Bouldin Index (cDBI) metric.

## 4. Conclusions

The authors investigated the question of choosing the optimal number of topics for building a topic model for a given corpus of texts. The study of this direction is relevant from the moment of discovery of the topic modeling technique. The result of this study was a technique that allows you to determine the optimal number of topics for a corpus of texts.

It should be said that the proposed method was experimentally confirmed under the following conditions:

- A small collection of documents;
- English language documents (monolingual text corpus);
- Thematic uniformity.

An important methodological trick of the authors is the preparation of a topic model using sequential regularization. In previous studies of this collection of documents [45], the authors obtained numerical estimates of the coefficients for the regularizing components of the topic model (*μ*).

When forming a collection of texts, conditions were set that limited the number of topics of scientific articles according to the topic rubrics to 10. The essence of the experiment was to confirm the selected number of topics using an optimization approach based on the quality metric developed by the authors of the topic model—Cosine Davies Bouldin Index (cDBI).

As a result, the experiment showed that the maximum value of the cDBI metric for test corpus is achieved with the average number of topics equal to 16 with standard deviation 2. The result was obtained with a large number of model training to eliminate the influence of the order of documents in the collection.

In conclusion, it is important to emphasize that this study can serve as a methodological groundwork for the creation of software frameworks and proposes support for solving one of the fundamental problems of semantic text processing: Determining the sense of a text fragment (article).

**Author Contributions:** Data curation, F.K. and A.S.; Methodology, F.K.; Project administration, A.S.; Supervision, F.K.; Writing—original draft, A.S.; Writing—review & editing, A.S.

**Funding:** This research received no external funding.

**Conflicts of Interest:** The authors declare no conflict of interest.

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
