# Peer review of "The Number of Topics Optimization: Clustering Approach"

_make, doi:10.3390/make1010025_

Round 1
Reviewer 1 Report
In general, the paper presented is very mature and complete research result.
Perhaps in the Introduction section, it would be better at the end to describe more clearly describe the state-of-the-art and contributions of the research proposed in the paper.
The Discussion and Conclusion would be strengthened by
a) referring back to the initial scientific (not practical) aims of the work and relevance to related scientific problem domain and
b) clearly explaining how, in the authors' view, the specific outcomes allow to influence the development and improvement of explored the scientific area(s) and the ultimate implications of such improvements.
Author Response
Dear Reviewer,
We would like to express our gratitude for your words of support regarding our study.
It has taken us more than six months to prepare this publication and your high evaluation helps us to successfully complete our work.
In accordance with your comments, the following adjustments have been made:
1. We have expanded the Conclusion and added a description of the scientific component of our result.
2. We strengthened the conclusions by adding our opinion on the application of the research results as well as more clearly formulated the conditions in which we conducted experimental confirmation of our methodology.
Respectfully,
The authors
Reviewer 2 Report
In this paper, the authors address the problem of determing the optimal number of topics. Therefore, the authors, after analysing the existing topic models propose a new one, but without comparing the results obtained with the corresponding ones for the classical topic models such as Coherence, Constrast, etc.
The citation order does not respect the MDPI journals requirements (see Authors Instructions). Also, it is not a good practice to say "the research [13] of 2018", maybe "a very recent research ..." (there are other phrases like this).
Some phrases (even in abstract) are hard to read (see "The cornerstone of the proposed new method of determining the 10 optimal number of topics based on the following principles ...") Also there are others English language problems such as "have been using successfully for clustering texts for many years" (have been used).
Author Response
Dear Reviewer,
We would like to express our gratitude for your words of support regarding our study.
It has taken us more than six months to prepare this publication and your high evaluation helps us to successfully complete our work.
In accordance with your comments, the following adjustments have been made:
1. We have changed the order of citing and design of the list of references in accordance with the requirements of MDPI journals,
2. We have corrected speech errors and improved the translation into English. For your convenience, corrections are highlighted in the text.
Respectfully,
The authors